# Constructing 3D Skeleton on Commercial Copper Foil via Electrophoretic Deposition of Lithiophilic Building Blocks for Stable Lithium Metal Anodes

**DOI:** 10.3390/nano13081400

**Published:** 2023-04-18

**Authors:** Yun Jiang, Wenqi Zhang, Yuyang Qi, Yuan Wang, Tianle Hu, Pengzhang Li, Chuanjin Tian, Weiwei Sun, Yumin Liu

**Affiliations:** 1Institute of New Energy Materials and Devices, School of Materials Science and Engineering, Jingdezhen Ceramic University, Jingdezhen 333403, China; 2Institute for Interdisciplinary Research (IIR), Jianghan University, Wuhan 430056, China; 3College of Aerospace Science and Engineering, National University of Defense Technology, Changsha 410073, China

**Keywords:** current collector, lithiophilic skeleton, lithium nucleation, electrophoretic deposition, lithium metal anodes

## Abstract

Lithium (Li) metal has been regarded as the "Holy Grail" of Li battery anodes thanks to its high theoretic specific capacity and low reduction potential, but uneven formation of Li dendrites and uncontrollable Li volume changes hinder the practical applications of Li metal anodes. A three-dimensional (3D) current collector is one of the promising strategies to address the above issues if it can be compatible with current industrialized process. Here, Au-decorated carbon nanotubes (Au@CNTs) are electrophoretically deposited on commercial Cu foil as a 3D lithiophilic skeleton to regulate Li deposition. The thickness of the as-prepared 3D skeleton can be accurately controlled by adjusting the deposition time. Benefitting from the reduced localized current density and improved Li affinity, the Au@CNTs-deposited Cu foil (Au@CNTs@Cu foil) achieves uniform Li nucleation and dendrite-free Li deposition. Compared with bare Cu foil and CNTs deposited Cu foil (CNTs@Cu foil), the Au@CNTs@Cu foil exhibits enhanced Coulombic efficiency and better cycling stability. In the full-cell configuration, the Au@CNTs@Cu foil with predeposited Li shows superior stability and rate performance. This work provides a facial strategy to directly construct a 3D skeleton on commercial Cu foils with lithiophilic building blocks for stable and practical Li metal anodes.

## 1. Introduction

The emerging market of electric vehicles, portable electronic devices, and grid-scale energy storage has stimulated intensive research into high-energy-density rechargeable batteries [1,2]. Metallic Li has been considered as the ultimate anode material to construct full cells with Li-free cathodes for next-generation energy storage devices, thanks to its high specific capacity of 3860 mA h g^−1^ and low electrochemical potential (−3.040 V versus the standard hydrogen electrode) [3]. However, the practical commercialization of Li metal anodes faces the challenges of safety concerns and low Coulombic efficiencies (CEs) owing to uncontrolled Li dendrite formation and relative infinite volume change during Li stripping/plating cycles [4,5]. To address these issues, considerable research efforts have been devoted in recent years, such as electrolyte engineering [6,7,8], artificial solid electrolyte interphase (SEI) layer construction [9,10,11], electrode design [12,13,14], separator modification [15,16,17], and so on. Among the abovementioned strategies, three-dimensional (3D) current collectors with a high specific surface area can not only reduce the localized current density, but also mitigate Li volume change during the charge/discharge process, which effectively suppresses the growth of Li dendrites and stabilizes the electrode dimension [18,19].

Recently, various advanced 3D current collectors have been developed to construct high-performance Li metal anodes [20]. Owing to the high conductivity and excellent electrochemical stability, commercial Ni and Cu foams have been widely studied as 3D current collectors [21,22,23]. However, these metallic foams with millimeter-scale thickness sacrifice a certain energy density of the battery. Besides, carbon-based materials, such as carbon nanotubes [24,25], carbon fibers [26,27], graphene [28,29], and hollow/porous carbons [30,31], can also be designed into different architectures with high conductivity, large surface area, and excellent stability for 3D current collectors. Nevertheless, there are still some challenges to be solved in practical application processes of these carbon-based 3D current collectors, including poor surface lithiophilicity, a complex fabrication process, and incompatibility with current battery manufacturing technology. Cu foil is the most commonly used anode current collector in the state-of-the-art architecture of Li-ion batteries, thus much effort has been devoted to modifying commercial Cu foil to guide dendrite-free Li deposition. For example, hollow carbon spheres with Au nanoparticles inside were coated on the surface of Cu foil to achieve selective deposition and stable encapsulation of Li metal, which eliminate the formation of Li dendrites and improve cycling performance [32]. In order to address the intrinsic lithiophobicity of Cu, various materials have been employed to modify the Cu foils, such as Cu_2_O [33], CuO [34], Cu_2_S [35], nitrogen-doped porous copper oxide nanosheet [36], Ag nanoparticles [37], ZnO nanorod arrays [38], MXene nanosheet arrays [39], benzotriazole [40], LiBi/LiAl/LiAu alloy [41], Li_4_Ti_5_O_12_ [42], and so on. Although these lithiophilic materials can reduce Li nucleation overpotential and regulate Li deposition behaviors, the finite surface area of the planar electrode still restricts the suppression effect to Li dendrite formation. Therefore, different 3D structures with a large surface area, including carbon-based 3D hosts [43,44], porous inverse opal Ni structures [45], and polymer nanofiber-based networks [46], have been designed and constructed on Cu foils, which significantly reduced the localized current density and homogenized Li ion flux. Despite these developments, there is still a lack of effective and facile strategies to construct lithiophilic 3D skeletons on commercial Cu foils.

Electrophoretic deposition (EPD) is a colloidal process wherein the suspended particles are forced to move toward an electrode by an electric field and deposited on this electrode, which has been developed to produce composite materials for coatings and films with manipulated structures [47]. EPD can be applied to any solid with certain particle surface charges in a stable colloidal suspension and is desirable for continuous and scalable production. In this work, we demonstrated a facial and effective strategy to construct 3D skeletons on commercial Cu foils with Au-decorated carbon nanotubes (Au@CNTs) as the lithiophilic building blocks. The as-synthesized Au@CNTs with controllable thickness were deposited on Cu foils via an EPD process. The obtained 3D conductive skeleton constructed by CNTs delivers a large surface area, which effectively reduces the localized current density and achieves homogeneous Li ion transport. Moreover, highly dispersed Au nanoparticles anchored on the CNTs not only improve the affinity with Li metal, but also enrich the nucleation sites for Li deposition. Thanks to these merits, the Au@CNTs-deposited Cu foil (Au@CNTs@Cu foil) shows a CE of 96.4% after 300 cycles at 1 mA cm^−2^ with 1 mA h cm^−2^ and 97.0% after 100 cycles at 2 mA cm^−2^ with 4 mA h cm^−2^, which are much higher than the values achieved by bare Cu foil and CNTs-deposited Cu foil (CNTs@Cu foil). The uniform and dendrite-free deposition of Li metal on Au@CNTs@Cu foil has been verified, guaranteeing superior stability and rate performance in full-cell configuration paired with LiFePO_4_ cathodes.

## 2. Materials and Methods

### 2.1. Electrophoretic Deposition of CNTs and Au@CNTs on Cu Foil

To synthesis Au-decorated carbon nanotubes (Au@CNTs), 50 mg of commercial CNTs was firstly ultrasonically dispersed into 50 mL ethanol, then 3 mL 3-Aminopropyltrimethoxysilane (APTMS) was added into the suspension and it was refluxed at 85 °C for 4 h, followed by washing with deionized (DI) water. The obtained APTMS-modified CNTs were redispersed into 100 mL DI water, then 100 μL 10 wt% HAuCl_4_ aqueous solution and 0.1 g trisodium citrate were added into the suspension. After stirring for 4 h at 85 °C, the Au@CNTs were obtained by centrifugation and washing with DI water, ethanol, and isopropanol, followed by ultrasonic dispersion into 200 mL isopropanol. To prepare the Au@CNTs suspension for electrophoretic deposition (EPD), 50 mg MgCl_2_∙6H_2_O was further added into the obtained Au@CNTs isopropanol solution. The CNTs suspension for EPD was prepared in the same way as the above process without the addition of HAuCl_4_ aqueous solution and trisodium citrate.

The commercial Cu foils were cut into small pieces with a size of 40 mm × 80 mm, then washed with acetone and ethanol several times before utilization. For the EPD process, a Cu foil coupled with an FTO glass were used as the electrodes. The Cu foil as the negative electrode and FTO glass as the positive electrode were placed 3 cm apart, and the DC voltage of 40 V was selected. In order to optimize the thickness of Au@CNTs on Cu foil, the deposition time for the EPD process was controlled to 10, 15, 20, and 30 min, respectively. The as-prepared Au@CNTs-deposited Cu foil (Au@CNTs@Cu foil) was washed with ethanol several times and dried in a vacuum oven at 45 °C for 2 h. Finally, the obtained Au@CNTs@Cu foil was cut into disks with a diameter of 14 mm as the current collector. To prepare CNTs@Cu foil, the CNTs suspension was used for the EPD process, the DC voltage of 40 V was selected, and the deposition time was set to 30 min.

### 2.2. Characterizations and Electrochemical Measurements

Scanning electron microscopy (SEM) images of all samples were characterized by field-emission SEM (FE-SEM, HITACHI SU8010). Transmission electron microscopy (TEM) images were observed by a JEM-2100F. Coin cells (CR2032) consisting of Cu foil, CNTs@Cu foil or Au@CNTs@Cu foil, and Li foil were assembled in an argon-filled glovebox. The electrolyte was composed of 1 M LiTFSI with 2% LiNO_3_ in DOL/DME (1:1 volume ratio). A Celgard membrane was employed as the separator for all cells. The assembled cells were cycled in a voltage range of 0 to 1.0 V at 0.05 mA cm^−2^ for the first five cycles. The CE at different current densities and capacities were measured by the LANHE test system (CT2001A). To evaluate the voltage hysteresis and cycling stability, Cu foil, CNTs@Cu foil, or Au@CNTs@Cu foil was predeposited with 4 mA h cm^−2^ of Li at a current density of 0.5 mA cm^−2^, followed by cycling at different current densities with a capacity of 1 mA h cm^−2^. In full-cell configuration, the predeposited Cu foil-Li, CNTs@Cu foil-Li, and Au@CNTs@Cu foil-Li were paired with LiFePO_4_ cathodes. The LiFePO_4_ cathodes were prepared by mixing active materials, carbon black, and polyvinylidene fluoride (7:2:1 by weight) in N-methyl-pyrrolidone (NMP) under stirring. The obtained homogeneous slurry was pasted on an Al foil and then transferred to a vacuum oven to dry at 120 °C overnight. The areal mass loading of LiFePO_4_ active materials was 5.5 mg cm^−2^. The liquid electrolyte was composed of 1.0 M LiPF_6_ in ethylene carbonate (EC)/ethylene methyl carbonate (EMC)/dimethyl carbonate (DMC) with a volume ratio of 1:1:1. Approximately 30 μL of electrolyte was dropped into each cell. Celgard membranes were used as the separators for all devices. Galvanostatic cycling was conducted on a LANHE test system (CT2001A) at room temperature.

## 3. Results and Discussion

Figure 1a illustrates the electrophoretic deposition (EPD) process of CNTs and Au@CNTs on commercial Cu foils. EPD is essentially a two-step process consisting of electrophoresis and deposition. Electrophoresis happens when the electric field is applied to the as-prepared CNTs or Au@CNTs suspension. MgCl_2_∙6H_2_O was added into the suspension to modify the negatively charged CNTs or Au@CNTs to positively charged Mg^2+^-CNTs or Mg^2+^-Au@CNTs for a cathodic EPD. The morphologies of the CNTs and as-synthesized Au@CNTs were characterized by scanning electron microscopy (SEM), as shown in Appendix A. The Au nanoparticles were decorated and homogeneously distributed on the surface of the CNTs, which was further confirmed by the transmission electron microscopy (TEM) image of the as-synthesized Au@CNTs, as shown in Figure 1d. After the EPD process, a 3D skeleton was constructed on a Cu foil by the CNTs or Au@CNTs building blocks, as shown in Appendix A and Figure 1b,c. This highly conductive 3D skeleton significantly reduces the localized current density and homogenizes charge flux compared with the pristine Cu foil. Nevertheless, the lithiophobic nature of CNTs shows a relatively low Li wettability and high Li nucleation barrier, which are not conducive to uniform Li deposition during long-term cycling. Noble metals, such as Ag, and Au, are often used to eliminate the Li nucleation batteries of the substrate for Li deposition, owing to their definite solubility in Li at room temperature and the formation of a solid solution surface layer [32]. The lithiophilic Au@CNTs building blocks improve the Li affinity of the obtained 3D skeleton and decrease the Li nucleation overpotential, which lead to more uniform Li deposition than that achieved in the case of CNTs@Cu foil, as shown in Figure 1a.

The EPD yield or the thickness of the deposited skeleton can be easily controlled by varying the EPD conditions, such as the suspension concentration, applied voltage, and deposition time. Herein, we employed the same suspension concentration and applied voltage for all of the samples. The thickness of the obtained skeletons was optimized by controlling the deposition time. As shown in Figure 2a–d, the thickness of the 3D skeletons based on Au@CNTs building blocks increased with the increase in deposition time; in the meantime, the color of the obtained Au@CNTs@Cu foil turned dark black. Figure 2e depicts the relationship between the thickness of the Au@CNTs-based 3D skeleton and deposition time. The thickness increases from ~3.3 μm at 10 min to ~14.4 μm at 30 min in a linear as well as the corresponding mass loading (Figure 2f). In order to determine the optimal deposition time, the obtained Au@CNTs@Cu foils with different thickness were paired with Li foils to assemble half cells for cycling tests. As shown in Figure 2g, the Coulombic efficiency (CE) of the cells based on the Au@CNTs@Cu foils with the deposition time of 10 min, 15 min, and 20 min drops remarkably after only 130, 150, and 175 cycles, respectively. The Au@CNTs@Cu foil—30 min exhibits the best cycling stability and shows a high CE of 96.4% after 300 cycles at 1 mA cm^−2^ with 1 mA h·cm^−2^. This result indicates that the thicker skeleton with a larger specific surface area can reduce the localized current density more effectively, leading to more homogeneous Li nucleation and deposition. The CNTs were also deposited on Cu foil for 30 min and the thickness of the CNTs-based 3D skeleton was approximately 15.0 μm (Appendix A), which is almost the same as that of the counterpart constructed by Au@CNTs. Therefore, the electrophoretic deposition time of all samples was set to 30 min in the following experiments.

To investigate Li plating/stripping behavior and long-term cycling stability, the half cells based on three different current collectors were assembled using Li foil as the counter electrode. As shown in Figure 3a, the Au@CNTs@Cu foil exhibits an average CE of 98.54% after 300 cycles at a current density of 1 mA cm^−2^ with an areal capacity of 1 mA h cm^−2^, whereas the CEs of the cells based on Cu foil and CNTs@Cu foil decay almost synchronously after only 100 cycles. The corresponding voltage profiles of Cu foil, CNTs@Cu foil, and Au@CNTs@Cu foil at different cycles are shown in Appendix A and Figure 3e,f. The current collector based on Au@CNTs@Cu foil shows excellent plating and stripping stabilities with the smallest voltage hysteresis of all three electrodes. Upon increasing the current density to 2 mA cm^−2^, an average CE of 97.83% was achieved after 180 cycles for the cell based on the Au@CNTs@Cu foil, which significantly outperforms the CE values achieved by Cu foil and CNTs@Cu foil (Figure 3b). When the areal capacity increased to 4 mA h cm^−2^, the average CE value of the Au@CNTs@Cu foil remained at 97.23% after 100 cycles at a current density of 2 mA cm^−2^. However, the CE values of the Cu foil and the CNTs@Cu foil both fluctuated with low retention, as shown in Figure 3c. It is worth noting that the cycling stability of the CNTs@Cu foil was obviously better than that of the Cu foil at a relatively high current density and large areal capacity (Figure 3b,c), which is more obvious than that at a low current density (Figure 3a). This result indicates that the 3D CNT skeleton constructed on Cu foil significantly reduced the localized current density during the Li plating/stripping process and homogenized charge flux. The overpotentials of the Li deposition on Cu foil, CNTs@Cu foil, and Au@CNTs@Cu foil are investigated as shown in Figure 3d. The overpotential of Li nucleation at the initial deposition stage can be determined by the difference between the bottom of the voltage drop and the latter voltage plateau, which is 288, 73, and 66 mV for Cu foil, CNTs@Cu foil, and Au@CNTs@Cu foil, respectively. Electrochemical impedance spectroscopy (EIS) spectra of different current collectors after cycling are shown in Appendix A. It can be indicated that Au@CNTs@Cu foil exhibits the most stable interfacial resistance of all samples. Benefitting from the improved Li affinity and decreased Li nucleation overpotential after the decoration of Au nanoparticles, the Au@CNTs@Cu foil exhibits the best cycling stability of all three current collectors.

In order to further evaluate the long-term cycling stability, 4 mA h cm^−2^ of Li was predeposited on Cu foil, CNTs@Cu foil, or Au@CNTs@Cu foil at a current density of 0.5 mA cm^−2^. Excessive Li is necessary during the evaluation of long cycle performance owing to the consumption of Li by the electrolyte [48]. Then, galvanostatic charge/discharge characterizations were carried out in Cu foil-Li||Li, CNTs@Cu foil-Li||Li, and Au@CNTs@Cu foil-Li||Li cells at different current densities with 1 mA h cm^−2^. As shown in Figure 3g, stable voltage profiles with a low overpotential of about 12 mV can be observed in the Au@CNTs@Cu foil-Li||Li cell for 1000 h, whereas the Cu foil-Li||Li and CNTs@Cu foil-Li||Li cells show increasing overpotentials for Li plating and stripping after only 536 and 760 h, respectively. The 536–546 h and 760–770 h intervals shown in the insets of Figure 3g reveal that the cell based on Au@CNTs@Cu foil-Li exhibits more stable voltage polarization that those achieved by the counterparts based on Cu foil-Li and CNTs@Cu foil-Li. As shown in Figure 3h, the overpotential became increasingly pronounced at a higher current density. The Au@CNTs@Cu foil-Li exhibits low overpotential values, which were 10, 13, 20, 25, and 45 mV at the current density of 0.5, 1, 2, 3, and 5 mA cm^−2^, respectively. In contrast, the Cu foil-Li and CNTs@Cu foil-Li show much higher voltage values with the increase in current density. These results demonstrate that Au@CNTs@Cu foil-Li achieves improved cycling stability, lower voltage polarization, and hysteresis compared with Cu foil-Li and CNTs@Cu foil-Li, which confirms the merits of the Au@CNTs@Cu foil as an advanced current collector for long-term cycling.

Morphology evolution of Li deposition on Cu foil, CNTs@Cu foil, and Au@CNTs@Cu foil was investigated to demonstrate the Li nucleation and growth on different substrates. In the case of Cu foil, the nucleation sites for Li deposition are randomly distributed owing to the lithiophobic nature of Cu substrate (Figure 4a). With increasing deposited Li content, the loose structure of metallic Li and nonuniform electric field distribution are easily formed, which enhance the tip effect and the growth of Li dendrites, as shown in Figure 4b. As for CNTs@Cu foil, the 3D skeleton significantly reduced the localized current density, leading to more uniform nuclei formation compared with the Cu foil (Figure 4c). According to the reported literature [49], the nuclei size is proportional to the inverse of overpotential. Therefore, the decreased overpotential, in the case of CNTs@Cu foil, induced a large nuclei size of deposited Li and inhibited the formation of Li dendrites to some extent, as shown in Figure 4d. However, many cracks and a rough surface can be observed in the top-view SEM image of Li-deposited CNTs@Cu foil, owing to insufficient Li affinity of the CNTs-based 3D skeleton. After the decoration of Au nanoparticles, the Au@CNTs@Cu foil achieved significantly improved Li affinity and decreased Li nucleation overpotential compared with the CNTs@Cu foil according to the above discussion. In addition, Au nanoparticles enriched the nucleation sites for Li deposition, which improved the Li nuclei density on Au@CNTs@Cu foil, as shown in the SEM image of deposited Li with 0.5 mA h cm^−2^ (Figure 4f). Furthermore, the highly distributed Au nanoparticles decorated on the surface of the CNTs can effectively induce metallic Li to deposit into the 3D skeleton, as illustrated in Figure 4e. Therefore, the surface morphology of Li deposited on Au@CNTs@Cu foil is much flatter and denser than that deposited on CNTs@Cu foil with the increase in plated Li content to 1 and 2 mA h cm^−2^, as shown in Figure 4f. Appendix A show the top-view SEM images of stripped CNTs@Cu foil and Au@CNTs@Cu foil after 50 cycles. It can be observed that there are residual dead Li particles and block mass on the surface of the CNTs@Cu foil, as shown in Appendix A, whereas there is no discernible dead Li on the Au@CNTs@Cu foil after continuous cycling (Appendix A). These results demonstrate that the 3D current collector based on Au@CNTs@Cu foil can effectively guide the deposition of metallic Li and inhibit the formation of Li dendrites.

To further verify the feasibility of Au@CNTs@Cu foil in practical applications, full cells were assembled by pairing LiFePO_4_ (LFP) cathodes with 4 mA h cm^−2^ of Li predeposited anodes (Cu foil-Li, CNTs@Cu foil-Li, and Au@CNTs@Cu foil-Li). As shown in Figure 5a, Au@CNTs@Cu foil-Li||LFP cell delivered specific capacities of ~156, 148, 140, 126, 109, and 98 mA h g^−1^ at a current rate of 0.2, 0.4, 1, 2, 4, and 6 C, respectively, which are the highest rate capacities among all devices. When the current rates returned to 0.2 C, the specific capacities were well recovered for all cells. However, the cell based on Au@CNTs@Cu foil-Li exhibited superior cycling stability than the counterparts based on Cu foil-Li and CNTs@Cu foil-Li in the next 50 cycles at 1 C. A capacity retention of 96.8% was achieved by Au@CNTs@Cu foil-Li||LFP, whereas the values of Cu foil-Li||LFP and CNTs@Cu foil-Li||LFP were only 70.0% and 92.7%, respectively. Furthermore, the Au@CNTs@Cu foil-Li||LFP cell shows less-polarized charge/discharge curves than the cells based on Cu foil-Li and CNTs@Cu foil-Li, as shown in Figure 5b and Appendix A. These results demonstrate that Au@CNTs@Cu foil-Li achieves stable Li plating/stripping behaviors and fast Li ion kinetics under high current rates, indicating the merits of the Au@CNTs@Cu foil current collector for practical applications in stable Li metal batteries.

## 4. Conclusions

In summary, we directly construct 3D skeletons on commercial Cu foils with Au@CNTs as the lithiophilic building blocks via a facile and effective electrophoretic deposition process. The obtained 3D conductive skeletons deliver a large specific surface area and reduced localized current density, achieving homogeneous Li ion flux. Highly dispersed Au nanoparticles decorated on the CNTs can sufficiently eliminate Li nucleation overpotential and improve Li affinity, leading to regulated Li nucleation and dendrite-free Li deposition. As a result, the Au@CNTs@Cu foil exhibited a significantly enhanced Coulombic efficiency and better cycling stability in half cells compared with Cu foil and CNTs@Cu foil. In full-cell configuration, the Au@CNTs@Cu foil-Li anode also demonstrates its excellent potential for practical applications. This study provides a feasible strategy to construct 3D current collectors with lithiophilic building blocks and promotes the development of stable Li metal anodes for industry.

## Figures and Tables

**Figure 1 nanomaterials-13-01400-f001:**
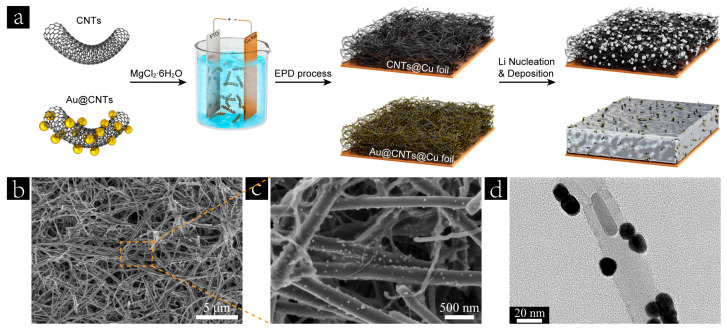
(**a**) Illustration of the EPD processes of CNTs and Au@CNTs on commercial Cu foils and corresponding Li nucleation/deposition processes. (**b**,**c**) Scanning electron microscopy (SEM) images of prepared Au@CNTs@Cu foil. (**d**) Transmission electron microscopy (TEM) image of the as-synthesized Au@CNTs.

**Figure 2 nanomaterials-13-01400-f002:**
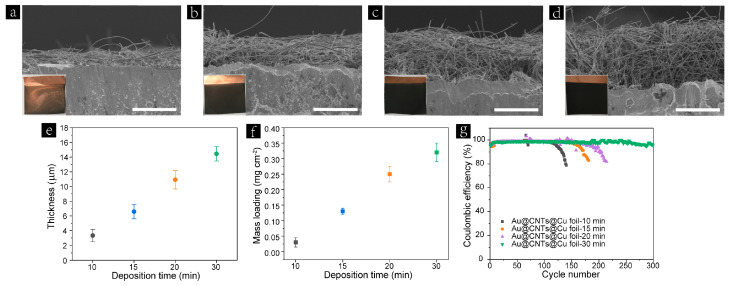
Cross-sectional SEM images of as-prepared Au@CNTs@Cu foil with different electrophoretic deposition times: (**a**) 10, (**b**) 15, (**c**) 20, and (**d**) 30 min. All scale bars are 10 μm. Insets show their corresponding digital photographs. (**e**) The thickness and (**f**) mass loading of as-prepared Au@CNTs@Cu foils at different electrophoretic deposition times. (**g**) Coulombic efficiency of the Au@CNTs@Cu foils with different thicknesses at a current density of 1 mA cm^−2^ with an areal capacity of 1 mA h cm^−2^.

**Figure 3 nanomaterials-13-01400-f003:**
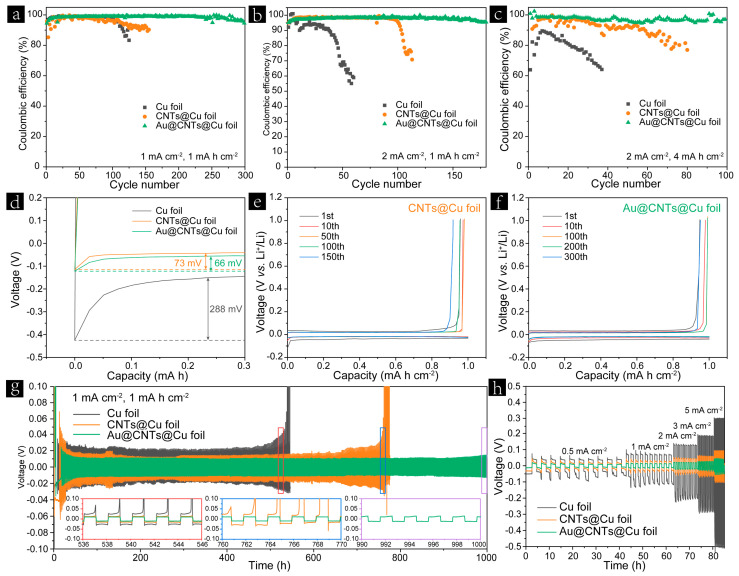
Coulombic efficiencies of Cu foil, CNTs@Cu foil, and Au@CNTs@Cu foil electrodes at (**a**) 1 mA cm^−2^ with 1 mA h cm^−2^, (**b**) 2 mA cm^−2^ with 1 mA h cm^−2^, and (**c**) 2 mA cm^−2^ with 4 mA h cm^−2^. (**d**) Voltage profiles of initial Li nucleation on Cu foil, CNTs@Cu foil, and Au@CNTs@Cu foil at a current density of 1 mA cm^−2^. Voltage profiles of (**e**) CNT@Cu foil and (**f**) Au@CNT@Cu foil under different cycles at a current density of 1 mA cm^−2^ with an areal capacity of 1 mA h cm^−2^. Galvanostatic cycling stabilities of the predeposited Cu foil-Li, CNTs@Cu foil-Li, and Au@CNTs@Cu foil-Li electrodes at (**g**) 1 mA cm^−2^ and (**h**) different current densities with an areal capacity of 1 mA h cm^−2^.

**Figure 4 nanomaterials-13-01400-f004:**
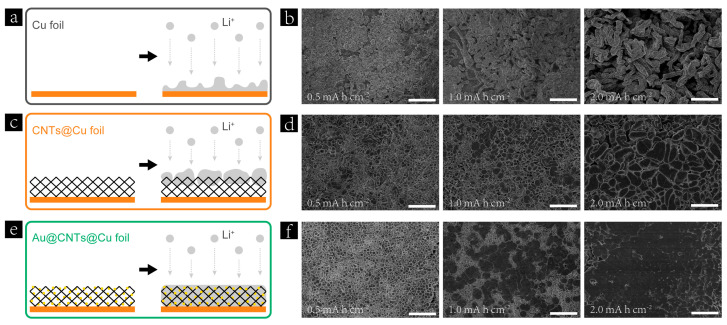
Schematic illustration of Li nucleation and deposition on (**a**) Cu foil, (**c**) CNTs@Cu foil, and (**e**) Au@CNTs@Cu foil. Top-view SEM images of (**b**) Cu foil, (**d**) CNTs@Cu foil, and (**f**) Au@CNTs@Cu foil after plating gradually increased areal capacities of 0.5, 1.0, and 2.0 mA h cm^−2^ at a current density of 1 mA cm^−2^. All scale bars are 25 μm.

**Figure 5 nanomaterials-13-01400-f005:**
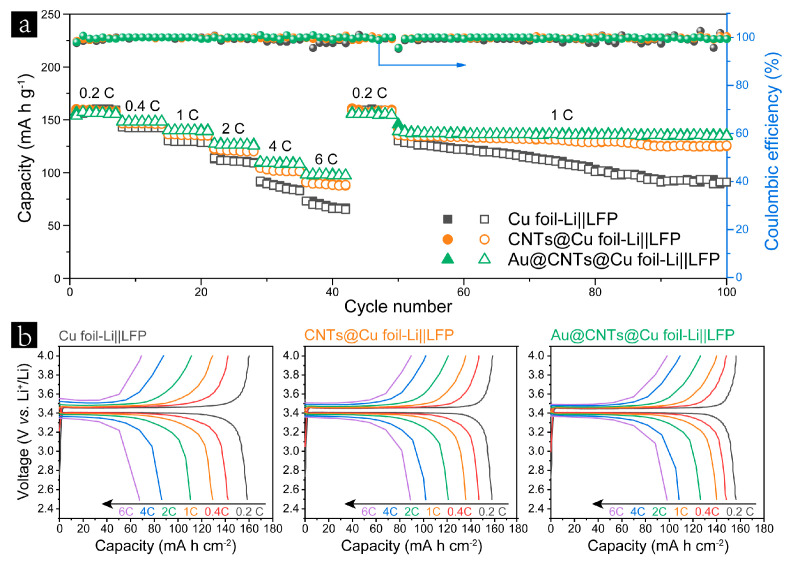
(**a**) Rate capabilities and cycling performances of the predeposited Cu foil-Li, CNTs@Cu foil-Li, and Au@CNTs@Cu foil-Li anodes in full-cell configuration at various rates from 0.2 to 6 C. (**b**) Charge/discharge curves of the full cells with the configuration of Cu foil-Li||LFP, CNTs@Cu foil-Li||LFP, and Au@CNTs@Cu foil-Li||LFP at different current rates from 0.2 to 6 C.

## Data Availability

Not applicable.

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
