# Peer review of "Constructing 3D Skeleton on Commercial Copper Foil via Electrophoretic Deposition of Lithiophilic Building Blocks for Stable Lithium Metal Anodes"

_nanomaterials, 2023, doi:10.3390/nano13081400_

Round 1
Reviewer 1 Report
This research paper reported synthesis of 3D skeletons on commercial Cu foils with Au@CNTs as the lithiophilic building blocks via a facile and effective electrophoretic deposition process. The obtained 3D conductive skeleton constructed by CNTs delivers large surface area, which effectively reduces the localized current density and achieves homogeneous Li ion transport. This paper can be accepted after addressing the following questions.
1. Why did the EPD process take up to 30 minutes? What if more than 30 minutes?
2. In this study, to prepare Au@CNTs suspension for electrophoretic deposition (EPD), 50 mg MgCl2∙6H2O were further added into the obtained Au@CNTs isopropanol solution. When Mg ions are deposited on the CNT, MgO crystals will exist. Could MgO crystals affect lithium deposition?
3. The numbers in the figures are too small.
4. Why does the voltage rise on one side when the cell is degraded when evaluating the symmetric cell in Figure 3g?
5. It would be good if EIS characteristic evaluation was added.
6. Some important papers on 3D nanostructured CNT electrode materials should be refered.
a. Three-dimensional macroporous CNTs microspheres highly loaded with NiCo2O4 hollow nanospheres showing excellent lithium-ion
storage performances. Carbon 128 (2018) 191-200.
b. Multiroom-structured multicomponent metal selenide–graphitic carbon–carbon nanotube hybrid microspheres as efficient anode materials for sodium-ion batteries. Nanoscale, 2018, 10, 8125.
Author Response
Journal: Nanomaterials
Manuscript ID: nanomaterials-2322010
Title: "Constructing 3D skeleton on commercial copper foil via electrophoretic deposition of lithiophilic building blocks for stable lithium metal anodes"
Author(s): Yun Jiang, Wenqi Zhang, Yuyang Qi, Yuan Wang, Tianle Hu, Pengzhang Li, Chuanjin Tian, Weiwei Sun, Yumin Liu
Dear Editor,
We thank the reviewer for his/her thoughtful comments and suggestions, and believe we have satisfactorily addressed them all in the revised manuscript. Below we reproduce the reviewer’ original comments and addressed the reviewer' concerns with a point-by-point format. The responses are embedded below each comment highlighted in yellow. The responses also indicate what changes have been made in the revised manuscript. We attach a revised manuscript where all the changes made are highlighted in yellow. We hope the paper is now acceptable for publication in Nanomaterials. Please feel free to let me know if there is any further question.
Sincerely,
Yumin Liu
Associate Researcher
Institute of New Energy Materials and Devices
School of Materials Science and Engineering
Jingdezhen Ceramic University
Jingdezhen 333403
China
E-mail: ymliu@jhun.edu.cn
A point-by-point response to reviewers’ comments is listed in the following:
Reviewer: 1
Comments:
This research paper reported synthesis of 3D skeletons on commercial Cu foils with Au@CNTs as the lithiophilic building blocks via a facile and effective electrophoretic deposition process. The obtained 3D conductive skeleton constructed by CNTs delivers large surface area, which effectively reduces the localized current density and achieves homogeneous Li ion transport. This paper can be accepted after addressing the following questions.
- Why did the EPD process take up to 30 minutes? What if more than 30 minutes?
Response:
Thanks to the reviewer’s comments. We have made a systematic study on the electrophoretic deposition time as shown in Figure 2. The experimental results show that the deposition time is directly proportional to the thickness of the obtained 3D skeletons. At the same time, we have tested the coulombic efficiency of current collectors with different thicknesses, and the performance of Au@CNTs@Cu foil-30 min is the best, so we set the electrophoretic deposition time to 30 minutes. When the deposition time continues to increase, the deposition thickness will further increase, which will lead to the weakening of adhesion.
- In this study, to prepare Au@CNTs suspension for electrophoretic deposition (EPD), 50 mg MgCl2∙6H2O were further added into the obtained Au@CNTs isopropanol solution. When Mg ions are deposited on the CNT, MgO crystals will exist. Could MgO crystals affect lithium deposition?
Response:
MgCl2∙6H2O is a commonly used suspension additive in electrophoretic deposition process, which is mainly used to modify the negatively charged CNTs or Au@CNTs to positively charged Mg2+-CNTs or Mg2+-Au@CNTs for a cathodic EPD and promote the directional movement of the deposit in the electric field. The reviewer’s suggestion is very constructive, whether Mg ions will be deposited together with carbon nanotubes or the influence on lithium metal deposition after deposition remains to be studied.
- The numbers in the figures are too mall.
Response:
Thanks for the reviewer’s suggestion, all numbers in the figures have been enlarged in our revised version, especially in Figure 2 and Figure 3.
- Why does the voltage rise on one side when the cell is degraded when evaluating the symmetric cell in Figure 3g?
Response:
The increase of voltage means that the uneven deposition of lithium metal produces dendrites, which leads to the short circuit of the symmetric cell.
- It would be good if EIS characteristic evaluation was added.
Response:
Thanks for the reviewer’s suggestion, EIS characteristic evaluation has been added in the revised supporting information, and corresponding discussions have also been highlighted in revised manuscript. Electrochemical impedance spectroscopy (EIS) spectra of different current collectors after cycling are shown in Figure S5. It can be indicated that Au@CNTs@Cu foil exhibits the most stable interfacial resistance of all samples.
Figure S5. The Nyquist plot of the electrochemical impedance spectroscopy (EIS) spectra of Cu foil, CNTs@Cu foil, and Au@CNTs@Cu foil before initial cycle and after 50 cycles at 1 mA cm−2 with 2 mAh cm−2.
- Some important papers on 3D nanostructured CNT electrode materials should be refered.
- Three-dimensional macroporous CNTs microspheres highly loaded with NiCo2O4 hollow nanospheres showing excellent lithium-ion
storage performances. Carbon 128 (2018) 191-200.
- Multiroom-structured multicomponent metal selenide–graphitic carbon–carbon nanotube hybrid microspheres as efficient anode materials for sodium-ion batteries. Nanoscale, 2018, 10, 8125.
Response:
The reviewer mentioned literatures have been cited in our revised version.

Reviewer 2 Report
Please, take care of grammatical errors, for example, in the abstract should read theretical instead of theoretic and electrophoretically instead of electrophoretic.
Is 3,860 mA h g-1 the correct value, please, check.
I do not see the importance of carbon nanotubes, other than it is difficult to make and that it enjoyes a fading popularity. Have the authors tried some simple carbon porous structures, for example. biochars?
Is gold the only lithiophilic alternative? Is there not a cheaper choice?
Author Response
Journal: Nanomaterials
Manuscript ID: nanomaterials-2322010
Title: "Constructing 3D skeleton on commercial copper foil via electrophoretic deposition of lithiophilic building blocks for stable lithium metal anodes"
Author(s): Yun Jiang, Wenqi Zhang, Yuyang Qi, Yuan Wang, Tianle Hu, Pengzhang Li, Chuanjin Tian, Weiwei Sun, Yumin Liu
Dear Editor,
We thank the reviewer for his/her thoughtful comments and suggestions, and believe we have satisfactorily addressed them all in the revised manuscript. Below we reproduce the reviewer’ original comments and addressed the reviewer' concerns with a point-by-point format. The responses are embedded below each comment highlighted in yellow. The responses also indicate what changes have been made in the revised manuscript. We attach a revised manuscript where all the changes made are highlighted in yellow. We hope the paper is now acceptable for publication in Nanomaterials. Please feel free to let me know if there is any further question.
Sincerely,
Yumin Liu
Associate Researcher
Institute of New Energy Materials and Devices
School of Materials Science and Engineering
Jingdezhen Ceramic University
Jingdezhen 333403
China
E-mail: ymliu@jhun.edu.cn
A point-by-point response to reviewers’ comments is listed in the following:
Reviewer: 2
Please, take care of grammatical errors, for example, in the abstract should read theretical instead of theoretic and electrophoretically instead of electrophoretic.
Response:
Thanks for the reviewer’s comments, grammatical errors in our manuscript have been revised. However, “theoretic” and “electrophoretic” in the abstract are the correction expression, which are consistent with the published literatures.
Is 3,860 mA h g-1 the correct value, please, check.
Response:
The theoretical specific capacity of Li metal is 3,860 mA h g-1, which has been checked.
I do not see the importance of carbon nanotubes, other than it is difficult to make and that it enjoyes a fading popularity. Have the authors tried some simple carbon porous structures, for example. biochars?
Response:
The 3D skeleton constructed by high conductive carbon nanotubes delivers large surface area, which effectively reduces the localized current density and achieves homogeneous Li ion transport. According to the reviewer’s suggestion, other porous carbon structures will also exhibit the same effect, if it can be deposited on the surface of copper foil by electrophoretic deposition.
Is gold the only lithiophilic alternative? Is there not a cheaper choice?
Response:
The reviewers' opinions are very constructive, and exploring other cheaper options is of great significance for reducing costs and expanding its application prospects.

Round 2
Reviewer 1 Report
I think the present form enable accept